# Artificial Intelligence in Automated Detection of Disinformation: A Thematic Analysis

**Fátima C. Carrilho Santos**

LabCom, University of Beira Interior, 6201-001 Covilhã, Portugal; fatima.catarina.santos@ubi.pt

**Abstract:** The increasing prevalence of disinformation has led to a growing interest in leveraging artificial intelligence (AI) for detecting and combating this phenomenon. This article presents a thematic analysis of the potential benefits of automated disinformation detection from the perspective of information sciences. The analysis covers a range of approaches, including fact checking, linguistic analysis, sentiment analysis, and the utilization of human-in-the-loop systems. Furthermore, the article explores how the combination of blockchain and AI technologies can be used to automate the process of disinformation detection. Ultimately, the article aims to consider the integration of AI into journalism and emphasizes the importance of ongoing collaboration between these fields to effectively combat the spread of disinformation. The article also addresses ethical considerations related to the use of AI in journalism, including concerns about privacy, transparency, and accountability.

**Keywords:** disinformation; artificial intelligence (AI); automatized detection; journalism

## 1. Introduction

The era of smartphones and social media has revolutionized the way content is transmitted and received, but not all of this content is accurate or truthful. The widespread use of the Internet and social media platforms has led to a significant increase in the prevalence of misleading information, known as disinformation. It is easily accessible and spreads rapidly in these digital environments.

In this article, we consider the definition of disinformation from the European Commission: "false, inaccurate, or misleading information designed, presented and promoted to intentionally cause public harm or for profit" (De Cock Buning 2018, p. 10). Disinformation, disguised as factual information, creates a distorted understanding of reality, leading to severe consequences for society by distorting people's perceptions of various issues. Political and health matters are particularly vulnerable to disinformation, as seen during the COVID-19 pandemic, during which false information spread rapidly through social media. However, disinformation can affect various areas by disseminating false knowledge about reality.

Although the phenomenon of disinformation gained significant attention during the 2016 US election campaigns, it has taken on a new dimension since 2020. The COVID-19 pandemic highlighted the critical importance of reliable and fact-based information for decision-making across all aspects of society, emphasizing its role as a fundamental pillar of democracy (Grizzle et al. 2021).

Disinformation not only poses challenges to discerning truthful information but also undermines journalistic credibility. It represents a significant obstacle to journalism as a means of knowledge production in society. Technological advancements, especially in artificial intelligence (AI), have not only increased the spread of disinformation but have also facilitated its automated creation and distribution. Consequently, the use of AI in disinformation creation and dissemination poses a significant challenge to the reliability of information, making it increasingly difficult to distinguish between facts and falsehoods.

AI can be defined as the process of automating tasks that typically require human intelligence. In other words, it involves replicating diverse facets of human thinking and

behavior (Russell and Norvig (2013). However, Bostrom (2018) emphasized that the most significant challenge for AI lies in attaining this behavior when performing tasks that necessitate common sense and comprehension of natural language, areas in which humans instinctively excel without conscious effort.

AI systems strive to replicate human-like reasoning, learning, planning, creativity, and numerous other capabilities. Algorithms, which comprise sequences of instructions or operations, are crafted to accomplish specific objectives. Present endeavors in the development of systems to identify misleading language in disinformation have emphasized automatic classification techniques and a range of algorithms (Bhutani et al. 2019).

One notable subfield of AI is machine learning, which enables computers to autonomously identify patterns in extensive datasets through algorithms without the need for explicit programming. Machine learning plays a vital role in tasks that require machines to learn from experience (Oliveira 2018). Consequently, it focuses on creating algorithms and procedures to enhance computers' performance in executing activities.

Furthermore, while AI can be used to amplify disinformation, it also plays a crucial role in identifying disinformation. According to Singh et al. (2021), advancements in this domain involve multimodal automated detection, which incorporates both textual and visual cues to identify disinformation. Various machine learning and classification algorithms have been employed to assign categories to specific datasets based on their characteristics.

Efforts have been consistently made to develop machine learning-based systems and natural language processing (NLP) algorithms for disinformation detection. However, thus far, there have been no flawless solutions. NLP, a subfield of AI, enables machines to comprehend, interpret, and replicate human language.

One challenge associated with utilizing AI and machine learning for disinformation detection constitutes ethical concerns. Biases and prejudices can infiltrate these systems, leading to erroneous outcomes. Moreover, ethical considerations pose obstacles to the use of AI in journalism, including the lack of monitoring and transparency and the potential suppression of creativity. Nevertheless, as stated by Jamil (2021), AI has been widely employed in journalism to automate repetitive tasks, particularly in data collection and identifying relevant patterns and trends for news reporting.

While this article examines different approaches to leverage AI in combating disinformation, the main focus is on automated fact-checking, which has been supported by existing research, such as Graves and Cherubini (2016) and Kertysova (2018). As we delve into the subsequent section on the intersection of AI and blockchain technologies, automated fact-checking, especially when combined with these two technologies, can encompass the three key stages of traditional fact-checking: identification, verification, and distribution (Graves 2018; Nakov et al. 2021). While other sections discuss different approaches, they can be integrated into an automated fact-checking system. However, a separate approach was chosen since automated fact-checking does not necessarily require the inclusion of all the other alternatives described.

## 2. Methods and Experimental Results

The objective of this study was to synthesize the current possibilities of using AI to combat disinformation. Disinformation is currently a growing problem, and it is increasingly important to understand how we can utilize AI to help address this issue.

Thematic analysis was chosen as the methodology for this study to provide a contextual understanding of the topic from the perspective of information sciences, distinguishing it from the more prevalent approaches in the literature, which often focus on specialized areas, such as computer engineering.

Thematic analysis, as a qualitative approach, allows for a comprehensive understanding of the potential of a given issue by gathering diverse aspects and data, and it facilitates the exploration of relationships between different concepts and interpretations (Alhojailan 2012). Therefore, our objective is to provide a comprehensive understanding of the current

state of the research regarding the utilization of AI to combat disinformation using this methodology.

To collect relevant material about the topic under investigation, we initially selected multiple databases, including Web of Science, Google Scholar, and IEEE Xplore. We specifically included IEEE Xplore to comprehend the current literature in the field of computer science, although our review primarily focused on collecting studies from the perspective of the social sciences.

To search for material related to the explored topic, we utilized keywords, such as "artificial intelligence" and "disinformation". However, due to the variety of terms used to refer to what is understood as disinformation in this context, we also included keywords such as "fake news" and "misinformation". The collected material covered the period between 2012 and 2023, prioritizing recent studies and research articles, while also including relevant book chapters.

Subsequently, we selected the most relevant material for the topic under study, particularly contextual approaches that provided a better understanding of the subject from the perspective of the social sciences. The data analysis was conducted systematically, involving the cross-referencing of ideas from various authors on specific subtopics within the main theme and employing cross-validation methods to ensure the accuracy and consistency of the information.

Employing thematic analysis, we conducted a literature review to identify key issues related to this topic, exploring the current possibilities of utilizing AI in combating disinformation. These possibilities include automated fact-checking systems, which may or may not incorporate automated sentiment analysis and which often involve automated text analysis. Additionally, we observed that the literature frequently mentions the convergence between automated systems and human knowledge in the verification process, as well as the combination of blockchain technology and AI.

## 3. Automated Fact-Checking for Disinformation Detection

In the context of disinformation, fact-checking refers to the process of verifying information to confirm the accuracy of fact-based statements. Fact-checking has always been part of the journalistic production routine, but with the rise of disinformation, journalists have taken on an even greater responsibility in deepening the practice of information verification. Consequently, various fact-checking agencies have emerged, with some linked to existing newsrooms and others operating independently in the market.

Since the 2000s, the number of projects related to fact-checking has continued to increase. According to Cazetta (2018), interest was sparked with the launch in 2003 of Factcheck.org, created by the journalist Brooks Jackson in the United States, with a focus on verifying statements made by public figures. Rosenstiel, in his book *The New Ethics of Journalism: Principles for the 21st Century* Rosenstiel (2013), argued that the fact-checking movement originated after the 1988 US election, when a *Washington Post* columnist, David Broder, classified candidates' statements in debates as "lies," "disqualified," and "demagogic," among other adjectives.

Some authors have also asserted that the initiative began in 1982, during the term of US President Ronald Reagan, when American journalists started to challenge inaccurate statements made by the president during press conferences (Cazetta 2018). However, in this case, it was primarily a contestation of a politician's statements and did not follow a structured methodology, unlike current fact-checking practices.

The proliferation of disinformation driven by the digital context has significantly boosted fact-checking efforts (do Nascimento 2021). In response, fact-checking projects aim to restore the credibility of journalism by exposing errors and incorrect information circulating on social media or in public figures' speeches. As Spinelli and Santos (2018) noted, what renders fact-checking a significant practice for journalism in the post-truth era is its emphasis on transparency and credibility.

Canavilhas and Ferrari (2018) pointed out that several renowned newspapers, such as the *Washington Post*, the *New York Times*, and *Le Monde*, have invested in these fact-checking formats, with the Americas and Europe being the continents with the most fact-checkers. Graves and Cherubini's study (2016) highlighted the growth of fact-checking sites throughout Europe, albeit with varying organizational forms and orientations, but with a shared commitment to publicly evaluate claims made by powerful actors, such as politicians, and in some cases information disseminated by the media.

Despite the growing interest in establishing fact-checking initiatives, it is noteworthy that the consumption of information generated by fact-checking agencies is often restricted to a specific audience, primarily journalists who rely on these verifications to validate their texts (do Nascimento 2021). According to Barrera et al. (2020), who studied the impact of content produced by fact-checking agencies compared to politicians' propaganda, there has been an absence of reaction from voters to fact-checking.

Experts and fact-checking organizations face the daunting task of investigating a vast amount of content, and AI has further intensified this workload, particularly due to the proliferation of bots deliberately generating and disseminating disinformation (Demartini et al. 2020). Due to several constraints surrounding fact-checking, such as time and resource limitations, new methods of fact-checking have been proposed, mainly based on the automatic detection of fake news. Thus, Jiang et al. (2021) argued that automated tools for detecting fake news, such as machine learning and specifically a type of machine learning algorithm, deep learning models, have become an essential requirement.

According to Kertysova's (2018) study, the initial proposals for automating online fact-checking emerged approximately a decade ago. However, the interest in research and investigation around AI-assisted fact-checking was triggered by Donald Trump's election as President of the United States, due to the large volume of erroneous information that was propagated: "The last few years have seen a wave of additional funding being earmarked for automated fact-checking initiatives that would help practitioners identify, verify, and correct social media content" (Kertysova 2018, p. 3).

One of the first challenges that had to be overcome initially in automated fact-checking was the lack of data to train the models. In 2017, online repositories with large volumes of information began to appear due to the growth of fact-checking sites. At the beginning of these developments, a database consisting of 106 fact checks from Politifact, one of the main fact-checking sites in the United States, was founded. Thus, a significant leap occurred in mid-2017 with the collection of a corpus of 12,800 checks from Politifact, and since then, the volume of databases has grown (García-Marín et al. 2022).

While creating extensive information repositories has been a significant challenge in AI-assisted fact-checking, ensuring data quality is equally crucial for designing effective algorithmic solutions to combat disinformation. Torabi Asr and Taboada (2019) argued that databases should be composed of samples of both false and true information in a balanced distribution across a range of topics. In a 2019 study, they reviewed available databases and presented a new repository called MisInfoText.

The primary objective of contemporary fact-checking endeavors is to ascertain the accuracy of information disseminated through social media channels (García-Marín et al. 2022). AI can automate the fact-checking process, especially through machine learning, NLP, and other sub-areas of AI.

There are also other automated alternatives for detecting disinformation, which can be part of fact-checking systems or exist individually. These alternatives include analyzing and verifying sources of information, cross-checking information, reviewing evidence, and more. Monitoring social media using AI-based recommendation systems, for example, can also be crucial in mitigating the spread of disinformation.

## 4. Detection of Disinformation through Language Analysis

Automated fact-checking can include text analysis performed by AI, which is useful in classifying news as true or false. In these cases, in addition to resources based on word

occurrences and word relationships (both semantic and syntactic), it is also necessary to have resources based on how humans perform fact-checking. That is, the automated classification of information examines human behavior in the process of manually detecting disinformation (Školkay and Filin 2019).

In fact-checking conducted by humans, contextual factors are considered, including the historical background, individuals involved, locations, and other relevant specifics related to the event. Thus, when automated detection of disinformation includes these characteristics, it is closer to human accuracy.

According to Kertysova (2018), automated technologies have limitations in evaluating individual statements. Current AI systems excel at identifying simple statements and assertions, but they struggle with more complex ones. The same limitation applies to expressions, in which context and culture are necessary.

Despite significant advancements in NLP techniques for automated text analysis, challenges remain in understanding fundamental human concepts, such as sarcasm or irony. AI-based systems currently struggle to address disinformation that relies on subtler forms of expression beyond explicit content. Additionally, linguistic barriers and specific cultural/political environments in each country pose even greater challenges.

A common approach to managing these challenges is to involve humans in the text analysis process, especially when regarding machine learning algorithm training (Školkay and Filin 2019). For example, in a fake news classification project, humans can flag a news story as false or true in the first phase. Then, the program can learn from these indications to assign characteristics and make classification decisions based on patterns identified in the news.

This approach, referred to as semi-supervised learning, offers the potential to enhance the accuracy of automated text analysis by combining the human ability to comprehend language nuances with the efficiency and scalability of automated text processing.

In machine learning systems, data representation greatly affects the accuracy of results, and the content shared by social media users being generally in unstructured forms makes this process difficult. Therefore, unstructured data extracted from social networks need to undergo transformation into a structured format using text mining methods. The problem of text mining can be defined as the extraction of meaningful, useful, and previously unknown information from textual data.

## 5. Detection of Disinformation through Sentiment Analysis

Sentiment analysis is a natural language processing technique that can also be used to detect disinformation. Although AI is not infallible, especially in the context of sentiment analysis, there have been significant improvements in this field because emotional appeal in fake news content differs from that in true news since messages with a strong emotional appeal can influence how content is consumed, processed, and shared by consumers (Paschen 2019). In fake news, the body of the news is usually more focused on specific negative emotions, such as disgust and anger, and less on positive emotions, such as joy.

In the study by Alonso et al. (2021), sentiment analysis (SA) was defined as a branch of Natural Language Processing (NLP), which is responsible for designing and implementing models, methods, and techniques to determine whether a text is composed of objective or subjective information and, in the latter case, to determine whether such information is expressed positively, neutrally, or negatively, as well as whether it is expressed strongly or weakly.

Computational techniques identify the characteristics of sentiments referred to as polarity, allowing for the classification of sentiments as positive, negative, or neutral. These techniques also identify specific emotions, such as sadness, anger, and more. While these computational techniques are applied to analyze text and images in news or content, their usage has been relatively limited in certain languages, such as Portuguese (Maia et al. 2021).

By analyzing the sentiment associated with news or content and determining whether there is emotional polarization, indications can be identified that the content is false; thus,

sentiment analysis can be a useful tool for detecting fake news, especially in automated fake news detection. Therefore, sentiment analysis, while it can exist independently, is also a tool that can be part of the automated fact-checking process.

Given that a majority of the subjective content shared by users on social media revolves around opinions, sentiment analysis is also referred to as Opinion Mining, as noted by Alonso et al. (2021). Shu et al. (2017) reported that SA should play a role in determining resource allocation based on posts since people express their emotions or opinions about fake news through social media posts, such as skeptical opinions or sensational reactions. Alonso et al. (2021) also referred to sentiment as one of the resources that can be extracted from text for fake news detection since conflicting sentiments between news disseminators can indicate a high probability of fake news.

## 6. Using Human-in-the-Loop AI Systems for Detecting Disinformation

As discussed throughout this article, considering the limitations of both AI and humans, it can be argued that combining the efforts of humans and AI is advantageous in combating disinformation. As Demartini et al. (2020) argued, human-in-the-loop AI (HAI) systems combine the best of both worlds, with humans filtering what will be implemented in practice by a machine that performs tasks based on databases and machine learning algorithms.

This model aims to harness AI's capability for large-scale processing, while utilizing human intelligence for complex tasks beyond AI's reach, such as language understanding and ensuring fairness and applicability within the system. One significant advantage of AI-based systems is their capacity to comprehensively analyze vast amounts of content, surpassing human capabilities (Babakar and Moy 2016).

Nevertheless, HAI systems encounter further challenges in achieving optimal functionality and generating valid results, including the inconsistency of data quality due to variations among participants involved in training the AI system. For example, using crowdsourcing, i.e., collective intelligence gathered in a complex task, to collect "labels" from online people using a platform is different from relying on experts.

Despite this limitation, since the number of available experts is usually limited, to obtain high-volume and quality labels, effective quality control mechanisms for crowdsourcing need to be developed. Another challenge that comes with HAI systems is the bias that contributing humans can create and/or amplify in the annotated data and, consequently, in the models learned with these labeled data (prejudices and stereotypes of contributing individuals can be reflected in the generated labels). As highlighted by Ghafourifar et al. (2021), if we aspire to create a potent and intelligent AI-based tool capable of detecting fake news, we must address our own biases and practice skepticism when it comes to consuming, sharing, and creating content on social media platforms and the Internet.

It is necessary to avoid biased models while leveraging human resources and filling in their gaps to better harness the scalability of AI-based methods and ob7tain the best of both worlds. Kertysova (2018) presented an opposite concept to human-in-the-loop AI, involving a different degree of human involvement in the decision-making process of an AI system, called humans "out of the loop" of AI systems. This approach is more common in systems that have been trained with a large volume of data and use sophisticated algorithms to make decisions, operating completely without direct human intervention.

Regarding humans being "out of the loop," the author noted that there are legal reasons why humans need to be kept informed about content moderation. According to a recent study funded by the European Science-Media Hub, limiting the automated enforcement of decisions about problems uncovered by AI is essential to ensure human agency and natural justice, including the right to appeal. This right does not prevent the suspension of bot accounts at scale, but it ensures proper auditing of the implemented system processes:

"The European data protection framework—which includes the General Data Protection Regulation (GDPR)—allows people to know how organizations are using their data as

well as to contest certain decisions made by algorithms. Because developers cannot explain how algorithms produce certain outcomes (see previous section), complaints relating to the GDPR have already been lodged, several organizations have been sanctioned, and more cases are expected to follow" (Kertysova 2018, p. 69).

### 7. The Application of AI Blockchain in the Automated Detection of Disinformation

In addition to AI, another technology that has been extensively discussed in the literature and could be useful in combating disinformation, especially when used in conjunction with AI, is blockchain.

Blockchain technology has been utilized for information validation and news aggregation. It is a decentralized record-keeping technology that allows for secure storage and validation of transactions to ensure that they cannot be altered. This technology, which is often used to create virtual currencies, can be leveraged to preserve and verify the integrity of news and other online content (Qayyum et al. 2019). Additionally, the authors concluded that blockchain technology, being a decentralized technology, promises to bring transparency and trust to journalism.

As do Val et al. (2021) explained, blockchain technology represents an alternative for verifying the content circulating on the Internet because, through it, users sign digital documents that are validated as true regarding a particular subject.

Therefore, technology tools based on blockchain can identify and compare false information with real facts. Several initiatives already employ blockchain to combat disinformation, including the News Provenance Project, Fake Check, and Democracy Notary. The *New York Times* also conducted a project to provide provenance metadata around news using blockchain technology to track the dissemination of news online and provide contextual information to news readers. There was also an effort to create initiatives around COVID-19, such as Hashlog.

The combination of AI and blockchain could lead to the creation of platforms with fewer limitations. Blockchain ensures secure and immutable data storage, while AI processes and analyzes large amounts of data in real time. One limitation of blockchain is its performance since it can only handle a limited number of transactions per second.

Thus, blockchain is used to store verified and trustworthy information about news sources, and AI, in turn, could be used to analyze and classify news in real time, comparing it with the information stored in the blockchain and identifying fake news.

However, in addition to detecting disinformation circulating on the Internet, Shae and Tsai (2019) pointed out that it is important to create mechanisms to minimize the impact of fake news before it is propagated; that is, it is necessary to create algorithms to predict fake news to anticipate the beginning of fake news propagation before it is actually spread.

This approach could be the most effective mechanism to combat the spread of fake news in the long run. Therefore, it is important and highly challenging to identify, label, and categorize different personal characteristics for diverse groups/communities and to develop corresponding intervention technologies. Personalized intervention technology can be developed by leveraging the AI-trustworthy news platform in blockchain.

To achieve this goal, some of the challenges include building databases of factual news, creating a trustworthy news blockchain, developing a blockchain crowdsourcing false news classification mechanism, developing scalable distributed intelligence contracting the blockchain network, and establishing a practical business model for building a trustworthy news ecosystem (Shae and Tsai 2019).

In conclusion, this study discusses a long-term strategy to develop models for predicting fake news before it spreads and to research the effectiveness of personalized intervention mechanisms. The combination of AI and blockchain technologies, such as in the practice of fact-checking, encompasses the three stages of fact-checking: identification, verification, and distribution (Graves 2018; Nakov et al. 2021). Westlund et al. (2022) pointed out that the identification stage gathers the highest concentration of technologies.

## 8. Discussion and Conclusions

Since the 2016 US presidential election, there has been growing interest in researching the use of new technologies, particularly AI, with a focus on automating fact-checking processes. In mid-2017, online repositories with large volumes of information started to emerge, effectively addressing the initial challenge of data scarcity for training automated fact-checking models. However, in addition to data quantity, ensuring the quality of the data used is crucial for designing effective algorithmic solutions to combat disinformation.

Currently, the main goal of fact-checking efforts is to verify the accuracy of information disseminated on social media, where AI has proved useful in automating the fact-checking process, particularly through machine learning, natural language processing, and other AI subfields.

In addition to automated fact-checking, which may also incorporate language analysis and sentiment analysis, the fight against disinformation requires a multifaceted approach that involves not only the use of AI and other technological tools but also human verification. Regarding other technologies, although this article primarily focused on artificial intelligence, it is worth mentioning that the combination of AI and blockchain technologies leads to a more efficient computer system for combating disinformation. AI can analyze and classify news in real time by comparing it with information stored on the blockchain, providing two-factor validation.

Therefore, it is important to acknowledge that automated detection of disinformation is only a part of the solution and should be complemented by other approaches that consider the complex nature of disinformation. Moreover, ethical concerns surrounding the use of AI in detecting disinformation must be addressed, including issues related to privacy, transparency, and accountability. Any use of AI in detecting disinformation must be guided by ethical principles and fundamental human values.

Efforts to combat disinformation are crucial for restoring the credibility of journalism, but a gap still exists between AI and journalistic practices. Although AI has been used to automate repetitive tasks and collect data for news, there remain ethical challenges and financial barriers that must be addressed to fully integrate AI into journalism. Hence, it is crucial to emphasize the importance of responsible implementation and the ongoing collaboration between technology and journalism.

**Funding:** This research received no external funding.

**Institutional Review Board Statement:** Not applicable.

**Informed Consent Statement:** Not applicable.

**Data Availability Statement:** Not applicable.

**Conflicts of Interest:** The author declares no conflict of interest.

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
