# Peer review of "Artificial Intelligence in Automated Detection of Disinformation: A Thematic Analysis"

_journalmedia, doi:10.3390/journalmedia4020043_

Round 1

Reviewer 1 Report

Undoubtedly, it is a paper focused on a relevant and timely issue: the use of artificial intelligence to combat disinformation. In the opinion of this reviewer, the paper could be published, but in its current version it presents several problems that should be corrected:

1. The introduction is too brief. It is suggested that a larger volume of papers on artificial intelligence and fact-checking be included. There is a lack of a real literature review and reference to other literature systematic reviews on the subject.

2. The method used is not clear. Further development is needed to explain the methodological process and provide information on key aspects to understand and evaluate the methodology, such as the number of papers analyzed, the inclusion and exclusion criteria, the type of studies (research articles, book chapters, doctoral theses, etc.) and the period analyzed. It is also suggested to provide access to the database of the sample.

3. In point 3, the part focused on fact-checking agencies should appear in the initial introductory section, since its content is not totally related (it is more general) to the object of study of the paper (automated fact-checking).

4. The discussion and conclusions are excessively brief. There is a lack of further development, especially in the section discussing the results of this study with other similar studies in order to compare the results obtained.

Author Response

All changes made, taking into account the notes made, are highlighted in red in the text. Please see the attachment.

Reviewer 2 Report

The authors must have defined a clear methodology for conducting a thematic analysis, the word context analysis does not make sense, in absence of a clear methodology. if the authors have written a review, must have made justice to the theme. Language could have been improved, several syntax errors could be avoided. 

It would have been written if the authors have incorporated a research design for conducting a review, including the search strategy, and inclusion and exclusion criteria. Total numbers of paper explored and incorporated after scanning, A PRISMA Chart could be provided. 

if the authors are ready to make such changes, the manuscript could be considered for publication. 

Author Response

(The authors gave the same response as above.)

Round 2

Reviewer 1 Report

After the changes made, the new version has improved considerably, so it could be published if the editors of the magazine consider it appropriate.

Author Response

Thank you.